# Model-Agnostic Private Learning

**Raef Bassily**[*]  **Om Thakkar**[†]  **Abhradeep Thakurta**[‡]

## Abstract

We design differentially private learning algorithms that are agnostic to the learning model assuming access to a limited amount of *unlabeled* public data. First, we provide a new differentially private algorithm for answering a sequence of $m$ online classification queries (given by a sequence of $m$ unlabeled public feature vectors) based on a private training set. Our algorithm follows the paradigm of subsample-and-aggregate, in which any generic non-private learner is trained on disjoint subsets of the private training set, and then for each classification query, the votes of the resulting classifiers ensemble are aggregated in a differentially private fashion. Our private aggregation is based on a novel combination of the distance-to-instability framework [26], and the sparse-vector technique [15, 18]. We show that our algorithm makes a conservative use of the privacy budget. In particular, if the underlying non-private learner yields a classification error of at most $\alpha \in (0,1)$, then our construction answers more queries, by at least a factor of $1/\alpha$ in some cases, than what is implied by a straightforward application of the advanced composition theorem for differential privacy. Next, we apply the knowledge transfer technique to construct a private learner that outputs a classifier, which can be used to answer an unlimited number of queries. In the PAC model, we analyze our construction and prove upper bounds on the sample complexity for both the realizable and the non-realizable cases. Similar to non-private sample complexity, our bounds are completely characterized by the VC dimension of the concept class.

## 1 Introduction

The main goal in the standard setting of differentially private learning is to design a differentially private learner that, given a private training set as input, outputs a model (or, a classifier) that is safe to publish. Despite being a natural way to define the private learning problem, there are several limitations with this standard approach. First, there are pessimistic lower bounds in various learning problems implying that the error associated with the final private model will generally have necessary dependence on the dimensionality of the model [2], or the size of the model class [9]. Second, this approach often requires non-trivial, white-box modification of the existing non-private learners [19, 11, 21, 26, 2, 27, 1], which can make some of these constructions less practical since they require making changes in the infrastructure of the existing systems. Third, designing algorithms for this setting often requires knowledge about the underlying structure of the learning problem, e.g., specific properties of the model class [4, 5, 9]; or convexity, compactness, and other geometric properties of the model space [2, 27].

We study the problem of differentially private learning when the learner has access to a limited amount of *public unlabeled* data. Our central goal is to characterize in a basic model, such as the standard PAC model, the improvements one can achieve for private learning in such a relaxed setting

---

[*]Department of Computer Science & Engineering, The Ohio State University. `bassily.1@osu.edu`

[†]Department of Computer Science, Boston University. `omthkkr@bu.edu`

[‡]Department of Computer Science, University of California Santa Cruz. `aguhatha@ucsc.edu`

compared to the aforementioned standard setting. Towards this goal, we first consider a simpler problem, namely, privately answering classification queries given by a sequence of *public unlabeled* data $\mathcal{Q} = \{x_1, \cdots, x_m\}$. In this problem, one is given a *private labeled* dataset denoted by $D$, and the goal is to design an $(\epsilon, \delta)$-differentially private algorithm that labels all $m$ public feature vectors in $\mathcal{Q}$. In designing such an algorithm, there are four main goals we aim to achieve: (i) We wish to provide an algorithm that enables answering as many classification queries as possible while ensuring $(\epsilon, \delta)$-differential privacy. This is a crucial property for the utility of such an algorithm since the utility in this problem is limited by the number of queries we can answer while satisfying the target privacy guarantee. (ii) We want to have a modular design paradigm in which the private algorithm can use any generic non-private algorithm (learner) in a *black-box* fashion, i.e., it only has oracle access to the non-private algorithm. This property is very attractive from a practical standpoint as the implementation of such an algorithm does not require changing the internal design of the existing non-private algorithms. (iii) We want to have a design paradigm that enables us to easily and formally transfer the accuracy guarantees of the underlying non-private algorithm into meaningful accuracy guarantees for the private algorithm. The most natural measure of accuracy in that setting would be the misclassification rate. (iv) We want to be able to use such an algorithm together with the public unlabeled data to construct a *differentially private learner* that outputs a classifier, which can then be used to answer as many classification queries as we wish. In particular, given the second goal above, the final private learner would be completely agnostic to the intricacies of the underlying non-private learner and its model. Namely, it would be oblivious to whether the model is simple logistic regression, or a multi-layer deep neural network.

Given the above goals, a natural framework to consider is *knowledge aggregation and transfer*, which is inspired by the early work of Breiman [8]. The general idea is to train a non-private learner on different subsamples from the private dataset to generate an ensemble of classifiers. The ensemble is collectively used in a differentially private manner to generate privatized labels for the given unlabeled public data. Finally, the public data together with the private labels are used to train a non-private learner, which produces a final classifier that is safe to publish.

**Related Work:** For private learning via knowledge aggregation and transfer, Hamm et al. [17] explored a similar technique, however their construction deviated from the above description. In particular, it was a white-box construction with weak accuracy guarantees; their guarantees also involved making strong assumptions about the learning model and the loss function used in training. In a recent work [23, 24], of which [24] is independent from our work, Papernot et al. gave algorithms that follow the knowledge transfer paradigm described above. Their constructions are black-box. However, only empirical evaluations are given for their constructions; no formal utility guarantees are provided. For the query-answering setting, a recent independent work [12] considers the problem of private prediction, but only in the single-query setting, whereas we study the multiple-query setting. The earliest idea of using ensemble classifiers to provide differentially private prediction can be traced to Dwork, Rothblum, and Thakurta from 2013.

**Our Techniques:** In this work, we give a new construction for privately answering classification queries that is based on a novel framework combining two special techniques in the literature of differential privacy, namely, the subsampling stability framework [22, 26] and the sparse vector technique [15, 18, 16]. Our construction also follows the knowledge aggregation and transfer paradigm, but it exploits the stability properties of good non-private learners in a quantifiable and formal manner. Our construction is based on the following idea: if a good learner is independently trained $k$ times on equally sized, independent training sets, then one would expect the corresponding output classifiers $h_1, \cdots, h_k$ to predict "similarly" on a new example from the same distribution. Using this idea, we show that among $m$ classification queries, one only needs to "pay the price of privacy" for the queries for which there is significant disagreement among the $k$ classifiers. Using our construction and the unlabeled public data, we also provide a final private learner. We show via formal and quantifiable guarantees that our construction achieves our four main goals stated earlier.

We note that our framework is not restricted to classification queries; it can be used for privately answering any sequence of online queries that satisfy certain stability properties in the sense of [26]. Due to space limitations, and to avoid distracting the reader from the main results of this work, we defer the description of the generic framework to the full version, and focus here on the special case of classification queries.

## 1.1 Our Contributions

**Answering online classification queries using the privacy budget conservatively:** In Section 3, we give our $(\epsilon, \delta)$-differentially private construction for answering a sequence of online classification queries. Our construction uses *any generic* non-private learner in a black-box fashion. The privacy guarantee is completely independent of the non-private learner and its accuracy. Moreover, the accuracy guarantee can be obtained directly from the accuracy of the non-private learner, i.e., the construction allows us to directly and formally "transform" the accuracy guarantee for the non-private learner into an accuracy guarantee for the final private algorithm.

We provide a new privacy analysis for the novel framework combining subsampling stability and sparse vector techniques. We analyze the accuracy of our algorithm in terms of its misclassification rate, defined as the ratio of misclassified queries to the total number of queries, in the standard (agnostic) PAC model. Our accuracy analysis is new, and is based on a simple counting argument. We consider both the realizable and non-realizable (agnostic) cases. In the realizable case, the underlying non-private learner is assumed to be a PAC learner for a hypothesis class $\mathcal{H}$ of VC-dimension $V$. The private training set consists of $n$ labeled examples, where the labels are generated by some unknown hypothesis $h^* \in \mathcal{H}$. The queries are given by a sequence of $m$ i.i.d. unlabeled domain points drawn from the same distribution as the domain points in the training set. We show that, with high probability, our private algorithm can answer up to $\approx n/V$ queries with a misclassification rate of $\approx V/n$, which is essentially the optimal misclassification rate attainable *without privacy*. Thus, answering those queries essentially comes with no cost for privacy. When answering $m > n/V$ queries, the misclassification rate is $\approx mV^2/n^2$. A straightforward application of the advanced composition theorem of differential privacy would have led to a misclassification rate $\approx \sqrt{m}V/n$, which can be significantly larger than our rate. This is because our construction pays a privacy cost only for "hard" queries for which the PAC learner tends to be incorrect. Our result for the realizable case is summarized below. We also provide an analogous statement for the non-realizable case (Theorem 3.5).

**Informal Theorem 1.1** (Corresponding to Theorem 3.4). *Given a PAC learner for a class $\mathcal{H}$ of VC-dimension $V$, a private training set of size $n$, and assuming realizability, our private construction (Algorithm 2) answers a sequence of up to $\tilde{\Omega}(n/V)$ binary classification queries such that, with high probability, the misclassification rate is $\tilde{O}(V/n)$. When the number of queries $m$ is beyond $\tilde{\Omega}(n/V)$, then with high probability, the misclassification rate is $\tilde{O}(mV^2/n^2)$.*

**A model-agnostic private learner with formal guarantees:** In Section 4, we use the knowledge transfer technique to bootstrap a private *learner* from our construction above. The idea is to use our private construction to label a sufficient number of public feature vectors. Then, we use these newly labeled public data for training a non-private learner to finally output a classifier. Since there is no privacy constraint associated with the public data, the overall construction remains private as differential privacy is closed under post-processing. Note that this construction also uses the non-private learner as a black box, and hence it is agnostic to the structure of such learner and the associated model. This general technique has also been adopted in [23]. Our main contribution here is that we provide formal and explicit utility guarantees for the final private learner in the standard (agnostic) PAC model. Our guarantees are in terms of upper bounds on the sample complexity in both realizable and non-realizable cases. Let $\mathrm{err}(h; \mathcal{D}) \triangleq \mathbb{P}_{(x,y)\sim\mathcal{D}} [h(x) \neq y]$ denote the true classification error of a hypothesis $h$. Given black-box access to an agnostic PAC learner for a class $\mathcal{H}$ of VC-dimension $V$, we obtain the following results:

**Informal Theorem 1.2** (Corresponding to Theorems 4.2, 4.3). *Let $0 < \alpha < 1$. Let $n$ be the size of the private training set. There exists an $(\epsilon, \delta)$-differentially private algorithm (Algorithm 3) that, given access to $m = \tilde{O}\left(\frac{V}{\alpha^2}\right)$ unlabeled public data points, w.h.p. outputs a classifier $\hat{h} \in \mathcal{H}$ such that the following guarantees hold: (i) Realizable case: $\mathrm{err}(\hat{h}; \mathcal{D}) \leq \alpha$ for $n = \tilde{O}\left(V^{3/2}/\alpha^{3/2}\right)$, and (ii) Agnostic case: $\mathrm{err}(\hat{h}; \mathcal{D}) \leq \alpha + O(\gamma)$ for $n = \tilde{O}\left(V^{3/2}/\alpha^{5/2}\right)$, where $\gamma = \min_{h\in\mathcal{H}} \mathrm{err}(h; \mathcal{D})$.*

Our bounds are only a factor of $\tilde{O}(\sqrt{V/\alpha})$ worse than the corresponding optimal non-private bounds. In the agnostic case, however, we note that the accuracy of the output hypothesis in our case has a suboptimal dependency (by a small constant factor) on $\gamma \triangleq \min_{h\in\mathcal{H}} \mathrm{err}(h; \mathcal{D})$.

We note that the same construction can serve as a private learner in a less restrictive setting where only the labels of the training set are considered private information. This setting is known as *label-private* learning, and it has been explored before in [10] and [6]. Both works have only considered pure, i.e., $(\epsilon, 0)$, differentially private learners, and their constructions are white-box, i.e., they do not allow for using a black-box non-private learner. The bounds in [10] involve smoothness assumptions on the underlying distribution. In [6], an upper bound on the sample complexity is derived for the realizable case. Their bound is a factor of $O(1/\alpha)$ worse than the optimal non-private bound for the realizable case.

## 2 Preliminaries

In this section, we formally define the notation, provide important definitions, and state the main existing results used in this work.

We denote the data universe by $U = \mathcal{X} \times \mathcal{Y}$, where $\mathcal{X}$ denotes abstract domain for unlabeled data (feature-vector space) and $\mathcal{Y} = \{0, 1\}$. An $n$-element dataset is denoted by $D = \{(x_1, y_1), (x_2, y_2), \ldots, (x_n, y_n)\} \in U^n$. For any two datasets $D, D' \in U^*$, we denote the symmetric difference between them by $D \Delta D'$.

We will use the standard notion of agnostic PAC learning [20] (see the full version for a definition). We will also use the following parameterized version of the definition of agnostic PAC learning.

**Definition 2.1** $((\alpha, \beta, n)$-learner for a class $\mathcal{H})$. *Let $\alpha, \beta \in (0, 1)$ and $n \in \mathbb{N}$. An algorithm $\Theta$ is an $(\alpha, \beta, n)$ (agnostic PAC) learner if, given an input dataset $D$ of $n$ i.i.d. examples from the underlying unknown distribution $\mathcal{D}$, with probability $1 - \beta$ it outputs a hypothesis $h_D$ with $\mathsf{err}(h_D; \mathcal{D}) \leq \gamma + \alpha$, where $\mathsf{err}(h; \mathcal{D}) \triangleq \mathbb{P}_{(x,y) \sim \mathcal{D}}[h_D(x) \neq y]$ and $\gamma \triangleq \min_{h \in \mathcal{H}} \mathsf{err}(h; \mathcal{D})$.*

Next, we define the notion of differential privacy.

**Definition 2.2** $((\epsilon, \delta)$-Differential Privacy [13, 14])**.** *A (randomized) algorithm $M$ with input domain $U^*$ and output range $\mathcal{R}$ is $(\epsilon, \delta)$-differentially private (DP) if for all pairs of datasets $D, D' \in U^*$ s.t. $|D \Delta D'| = 1$, and every measurable $S \subseteq \mathcal{R}$, we have that: $\Pr(M(D) \in S) \leq e^\epsilon \cdot \Pr(M(D') \in S) + \delta$, where the probability is over the coin flips of $M$.*

### 2.1 Distance to Instability Framework

Next, we describe the distance to instability framework from [26] that releases the exact value of a function on a dataset while preserving differential privacy, provided the function is sufficiently *stable* on the dataset. We define the notion of stability first, and provide a pseudocode for a private estimator for any function via this framework in Algorithm 1.

**Definition 2.3** ($k$-stability [26])**.** *A function $f : U^* \to \mathcal{R}$ is $k$-stable on dataset $D$ if adding or removing any $k$ elements from $D$ does not change the value of $f$, i.e., $\forall D' s.t. |D \Delta D'| \leq k$, we have $f(D) = f(D')$. We say $f$ is* stable *on $D$ if it is (at least) 1-stable on $D$, and* unstable *otherwise.*

The *distance to instability* of a dataset $D \in U^*$ with respect to a function $f$ is the number of elements that must be added to or removed from $D$ to reach a dataset that is not stable.

---

**Algorithm 1** $\mathcal{A}_{\mathsf{stab}}$ [26]: Private release of a classification query via distance to instability

---

**Input:** Dataset $D \in U^*$, a function $f : U^* \to \mathcal{R}$ for some range $\mathcal{R}$, distance to instability function associated with $f$: $\mathsf{dist}_f : U^* \to \mathbb{R}$, threshold: $\Gamma$, privacy parameter $\epsilon > 0$
1: $\widehat{\mathsf{dist}} \leftarrow \mathsf{dist}_f(D) + \mathsf{Lap}(1/\epsilon)$
2: **If** $\widehat{\mathsf{dist}} > \Gamma$, **then** output $f(D)$, **else** output $\perp$

---

**Theorem 2.4** (Privacy guarantee for $\mathcal{A}_{\mathsf{stab}}$)**.** *If the threshold $\Gamma = \log(1/\delta)/\epsilon$, and the distance to instability function $dist_f(D) = \arg\max_k [f(D)$ is $k$-stable], then $\mathcal{A}_{\mathsf{stab}}$ (Algorithm 1) is $(\epsilon, \delta)$-DP.*

**Theorem 2.5** (Utility guarantee for $\mathcal{A}_{\mathsf{stab}}$)**.** *If the threshold $\Gamma = \log(1/\delta)/\epsilon$, the distance to instability function is chosen as in Theorem 2.4, and $f(D)$ is $((\log(1/\delta) + \log(1/\beta))/\epsilon)$-stable, then Algorithm 1 outputs $f(D)$ with probability at least $1 - \beta$.*

The proof of the above two theorems follows from [26, Proposition 3].

# 3 Privately Answering Classification Queries

In this section, we instantiate the distance to instability framework (Algorithm 1) with the sub-sample and aggregate framework [22, 26], and then combine it with the sparse vector technique [15, 16] to obtain a construction for privately answering classification queries with a conservative use of the privacy budget (Algorithm 2 below). We consider here the case of binary classification for simplicity. However, we note that one can easily extend the construction (and obtain analogous guarantees) for multi-class classification.

**Note:** The full version [3] contains a detailed discussion of a more general framework together with a more modular and generic description of the algorithmic techniques. The description of the algorithms in this short version involves only classification queries.

A private training set, denoted by $D$, is a set of $n$ private binary-labeled data points $\{(x_1, y_1), \ldots, (x_n, y_n)\} \in (\mathcal{X} \times \mathcal{Y})^n$ drawn i.i.d. from some (arbitrary unknown) distribution $\mathcal{D}$ over $\mathcal{X} \times \mathcal{Y}$. We will refer to the induced marginal distribution over $\mathcal{X}$ as $\mathcal{D}_{\mathcal{X}}$. We consider a sequence of (online) classification queries defined by a sequence of $m$ unlabeled points from $\mathcal{X}$ $\mathcal{Q} = \{\tilde{x}_1, \cdots, \tilde{x}_m\} \in \mathcal{X}^m$, drawn i.i.d. from $\mathcal{D}_{\mathcal{X}}$, and let $\{\tilde{y}_1, \cdots, \tilde{y}_m\} \in \{0, 1\}^m$ be the corresponding true unknown labels. Algorithm 2 has oracle access to a non-private learner $\Theta$ for a hypothesis class $\mathcal{H}$. We will consider both realizable and non-realizable cases of the standard PAC model. In particular, $\Theta$ is assumed to be an (agnostic) PAC learner for $\mathcal{H}$.

---

**Algorithm 2** $\mathcal{A}_{\mathsf{binClas}}$: Private Online Binary Classification via subsample and aggregate; and sparse vector

---

**Input:** Private dataset: $D$, sequence of online unlabeled public data (defining the classification queries) $\mathcal{Q} = \{\tilde{x}_1, \cdots, \tilde{x}_m\}$, oracle access to a non-private learner $\Theta : U^* \to \mathcal{H}$ for a hypothesis class $\mathcal{H}$, cutoff parameter: $T$, privacy parameters $\epsilon, \delta > 0$, failure probability: $\beta$

1: $c \leftarrow 0$, $\lambda \leftarrow \sqrt{32 T \log(2/\delta)}/\epsilon$, and $k \leftarrow 34\sqrt{2}\lambda \cdot \log(4mT/\min(\delta, \beta/2))$
2: $w \leftarrow 2\lambda \cdot \log(2m/\delta)$, and $\widehat{w} \leftarrow w + \mathsf{Lap}(\lambda)$
3: Arbitrarily split $D$ into $k$ non-overlapping chunks of size $n/k$. Call them $D_1, \cdots, D_k$
4: **for** $j \in [k]$, train $\Theta$ on $D_j$ to get a classifier $h_j \in \mathcal{H}$
5: **for** $i \in [m]$ and $c \leq T$ **do**
6:      Let $\mathcal{S}_i = \{h_1(x_i), \cdots, h_k(x_i)\}$, and for $y \in \{0, 1\}$, let $\mathsf{ct}(y) = \#$ times $y$ appears in $\mathcal{S}_i$
7:      $\widehat{q}_{x_i}(D) \leftarrow \arg \max_{y \in \{0,1\}} [\mathsf{ct}(y)]$, $\mathsf{dist}_{\widehat{q}_{x_i}} \leftarrow \max\{0, \mathsf{ct}(\widehat{q}_{x_i}) - \mathsf{ct}(1 - \widehat{q}_{x_i}) - 1\}$
8:      $\mathsf{out}_i \leftarrow \mathcal{A}_{\mathsf{stab}}\left(D, \widehat{q}_{x_i}, \mathsf{dist}_{\widehat{q}_{x_i}}, \Gamma = \widehat{w}, \epsilon = 1/2\lambda\right)$
9:      **If** $\mathsf{out}_i = \bot$, **then** $c \leftarrow c + 1$ and $\widehat{w} \leftarrow w + \mathsf{Lap}(\lambda)$
10:      Output $\mathsf{out}_i$

---

**Theorem 3.1** (Privacy guarantee for $\mathcal{A}_{\mathsf{binClas}}$). *Algorithm $\mathcal{A}_{\mathsf{binClas}}$ (Algorithm 2) is $(\epsilon, \delta)$-DP.*

The proof of this theorem follows from combining the guarantees of the distance to instability framework [26], and the sparse vector technique [16]. The idea is that in each round of query response, if the algorithm outputs a label in $\{0, 1\}$, then there is "no loss in privacy" in terms of $\epsilon$ (as there is sufficient consensus). However, when the output is $\bot$, there is a loss of privacy. This argument is formalized via the distance to instability framework. Sparse vector helps account for the privacy loss across all the $m$ queries. A formal proof of this theorem is deferred to the full version.

**Theorem 3.2.** *Let $\alpha, \beta \in (0, 1)$, and $\gamma \triangleq \min_{h \in \mathcal{H}} \mathsf{err}(h; \mathcal{D})$. (Note that in the realizable case $\gamma = 0$). In Algorithm $\mathcal{A}_{\mathsf{binClas}}$ (Algorithm 2), suppose we set the cutoff parameter as $T = 3\left((\gamma + \alpha)m + \sqrt{(\gamma + \alpha)m \log(m/\beta)/2}\right)$. If $\Theta$ is an $(\alpha, \beta/k, n/k)$-agnostic PAC learner (Definition 2.1), where $k$ is as defined in $\mathcal{A}_{\mathsf{binClas}}$, then i) with probability at least $1 - 2\beta$, $\mathcal{A}_{\mathsf{binClas}}$ does not halt before answering all the $m$ queries in $\mathcal{Q}$, and outputs $\bot$ for at most $T$ queries; and ii) the misclassification rate of $\mathcal{A}_{\mathsf{binClas}}$ is at most $T/m = O(\gamma + \alpha)$.*

*Proof.* First, notice that $\Theta$ is an $(\alpha, \beta/k, n/k)$-agnostic PAC learner, hence w.p. $\geq 1 - \beta$, the misclassification rate of $h_j$ for all $j \in [k]$ is at most $\gamma + \alpha$. So, by the standard Chernoff's bound, with probability at least $1 - \beta$ none of the $h_j$'s misclassify more than $(\gamma + \alpha)m + $

$\sqrt{(\gamma + \alpha)m \log(m/\beta)/2} \triangleq B$ queries in $\mathcal{Q}$. Now, we use the following Markov-style counting argument (Lemma 3.3) to bound the number of queries for which at least $k/3$ classifiers in the ensemble $\{h_1, \ldots, h_k\}$ result in a misclassification.

**Lemma 3.3.** *Consider a set of* $\{(\tilde{x}_1, \tilde{y}_1), \ldots, (\tilde{x}_m, \tilde{y}_m)\} \subset \mathcal{X} \times \mathcal{Y}$*, and* $k$ *binary classifiers* $h_1, \ldots, h_k$*, where each classifier is guaranteed to make at most* $B$ *mistakes in predicting the* $m$ *labels* $\{\tilde{y}_1, \ldots, \tilde{y}_m\}$*. For any* $\xi \in (0, 1/2]$*,* $\left| \left\{ i \in [m] : |\{j \in [k] : h_j(\tilde{x}_i) \neq \tilde{y}_i\}| > \xi k \right\} \right| < B/\xi$*.*

Therefore, there are at most $3B$ queries $\tilde{x}_i \in \mathcal{Q}$, where the votes of the ensemble $\{h_1(\tilde{x}_i), \ldots, h_k(\tilde{x}_i)\}$ has number of ones (or, zeros) $> k/3$ (i.e., they significantly disagree). Now, to prove part (i) of the theorem, observe that to satisfy the distance to instability condition (in Theorem 2.5) for the remaining $m - 3B$ queries, it would suffice to have $k/3 \geq 32 \log\left(4mT/\min(\delta, \beta/2)\right)\sqrt{2T \log(2/\delta)}/\epsilon$ (taking into account the noise in the threshold passed to $\mathcal{A}_{\mathsf{stab}}$ in Step 8 of $\mathcal{A}_{\mathsf{binClas}}$[4]). This condition on $k$ is satisfied by the setting of $k$ in $\mathcal{A}_{\mathsf{binClas}}$. For part (ii), note that by the same lemma above, w.p. $1 - \beta$, there are at least $2k/3$ classifiers that output the correct label in each of the remaining $m - 3B$ queries. Hence, w.p. $\geq 1 - 2\beta$, Algorithm $\mathcal{A}_{\mathsf{binClas}}$ will correctly classify such queries. This completes the proof. $\qquad\square$

**Remark 1.** *A natural question for using Theorem 3.2 in the agnostic case is that how would one know the value of* $\gamma$ *in practice, in order to set the right value for* $T$*? One simple approach is to set aside half the training dataset, and compute the empirical misclassification rate with differential privacy to get a sufficiently accurate estimate for* $\gamma + \alpha$ *(as in standard validation techniques [25]), and use it to set* $T$*. Since the sensitivity of misclassification rate is small, the amount of noise added would not affect the accuracy of the estimation. Furthermore, with a large enough training dataset, the asymptotics of Theorem 3.2 would not change either.*

**Explicit misclassification rate:** In Theorem 3.2, it might seem that there is a circular dependency of the following terms: $T \rightarrow \alpha \rightarrow k \rightarrow T$. However, the number of independent relations is equal to the number of parameters, and hence, we can set them meaningfully to obtain non-trivial misclassification rates. We now obtain an explicit misclassification rate for $\mathcal{A}_{\mathsf{binClas}}$ in terms of the VC-dimension of $\mathcal{H}$. Let $V$ denote the VC-dimension of $\mathcal{H}$. First, we consider the realizable case ($\gamma = 0$). Our result for this case is formally stated in the following theorem.

**Theorem 3.4** (Misclassification rate in the realizable case). *For any* $\beta \in (0, 1)$*, there exists* $M = \tilde{\Omega}(\epsilon\, n/V)$*, and a setting for* $T = \tilde{O}\left(\bar{m}^2\, V^2/\epsilon^2\, n^2\right)$*, where* $\bar{m} \triangleq \max(M, m)$*, such that w.p.* $\geq 1 - \beta$*,* $\mathcal{A}_{\mathsf{binClas}}$ *yields the following misclassification rate: (i)* $\tilde{O}(V/\epsilon\, n)$ *for up to* $M$ *queries, and (ii)* $\tilde{O}\left(mV^2/\epsilon^2\, n^2\right)$ *for* $m > M$ *queries.*

*Proof.* By standard uniform convergence arguments [25], there is an $(\alpha, \beta, n/k)$-PAC learner with misclassification rate $\alpha = \tilde{O}\left(kV/n\right)$. Setting $T$ as in Theorem 3.2 with the aforementioned setting of $\alpha$, and setting $k$ as in Algorithm $\mathcal{A}_{\mathsf{binClas}}$ gives the setting of $T$ in the theorem statement. For up to $m = \tilde{\Omega}(\epsilon\, n/V)$ queries, the setting of $T$ becomes $T = O(1)$, and hence Theorem 3.2 implies $\mathcal{A}_{\mathsf{binClas}}$ yields a misclassification rate $\tilde{O}(V/\epsilon\, n)$, which is essentially the same as the optimal non-private rate. Beyond $\tilde{\Omega}(\epsilon\, n/V)$ queries, $T = \tilde{O}(m^2\, V^2/\epsilon^2\, n^2)$, and hence, Theorem 3.2 implies that the misclassification rate of $\mathcal{A}_{\mathsf{binClas}}$ is $\tilde{O}\left(mV^2/\epsilon^2\, n^2\right)$. $\qquad\square$

We note that the attainable misclassification rate is significantly smaller than the rate of $\tilde{O}\left(\sqrt{m}V/\epsilon\, n\right)$ implied by a direct application of the advanced composition theorem of differential privacy. Next, we provide analogous statement for the non-realizable case ($\gamma > 0$).

**Theorem 3.5** (Misclassification rate in the non-realizable case)**.** *For any* $\beta \in (0, 1)$*, there exists* $M = \tilde{\Omega}\left(\min\left\{1/\gamma, \sqrt{\epsilon\, n/V}\right\}\right)$*, and a setting for* $T = O(\bar{m}\gamma) + \tilde{O}\left(\bar{m}^{4/3}\, V^{2/3}/\epsilon^{2/3}\, n^{2/3}\right)$*, where* $\bar{m} \triangleq \max\{M, m\}$*, such that w.p.* $\geq 1 - \beta$*,* $\mathcal{A}_{\mathsf{binClas}}$ *yields the following misclassification rate: (i)* $O(\gamma) + \tilde{O}\left(\sqrt{V/\epsilon\, n}\right)$ *for up to* $M$ *queries, and (ii)* $O(\gamma) + \tilde{O}\left(m^{1/3}\, V^{2/3}/\epsilon^{2/3}\, n^{2/3}\right)$ *for* $m > M$ *queries.*

*Proof.* Again, by a standard argument, $\Theta$ is $(\alpha, \beta, n/k)$-agnostic PAC learner with $\alpha = \tilde{O}\left(\sqrt{kV/n}\right)$, and hence, it has a misclassification rate of $\approx \gamma + \tilde{O}\left(\sqrt{kV/n}\right)$ when trained on a dataset of size $n/k$. Setting $T$ as in Theorem 3.2 with this value of $\alpha$, and setting $k$ as in $\mathcal{A}_{\mathsf{binClas}}$, and then solving for $T$ in the resulting expression, we get the setting of $T$ as in the theorem statement (it would help here to consider the cases where $\gamma > \alpha$ and $\gamma \leq \alpha$ separately). For up to $m = \tilde{\Omega}\left(\min\left\{1/\gamma, \sqrt{\epsilon\, n/V}\right\}\right)$ queries, the setting of $T$ becomes $T = O(1)$, and hence Theorem 3.2 implies $\mathcal{A}_{\mathsf{binClas}}$ yields a misclassification rate $O(\gamma) + \tilde{O}\left(\sqrt{V/\epsilon\, n}\right)$, which is essentially the same as the optimal non-private rate. Beyond $\tilde{\Omega}\left(\min\left\{1/\gamma, \sqrt{\epsilon\, n/V}\right\}\right)$ queries, $T = O(m\gamma) + \tilde{O}\left(m^{4/3}\, V^{2/3}/\epsilon^{2/3}\, n^{2/3}\right)$, and hence, Theorem 3.2 implies that the misclassification rate of $\mathcal{A}_{\mathsf{binClas}}$ is $O(\gamma) + \tilde{O}\left(m^{1/3}\, V^{2/3}/\epsilon^{2/3}\, n^{2/3}\right)$. $\qquad\square$

## 4 From Answering Queries to Model-agnostic Private Learning

In this section, we build on our algorithm and results in Section 3 to achieve a stronger objective. In particular, we bootstrap from our previous algorithm an $(\epsilon, \delta)$-differentially private learner that publishes a final classifier. The idea is based on a knowledge transfer technique: we use our private construction above to generate labels for sufficient number of unlabeled domain points. Then, we use the resulting labeled set as a new training set for any standard (non-private) learner, which in turn outputs a classifier. We prove explicit sample complexity bounds for the final private learner in both PAC and agnostic PAC settings.

Our final construction can also be viewed as a private learner in the less restrictive setting of label-private learning where the learner is only required to protect the privacy of the labels in the training set. Note that any construction for our original setting can be used as a label-private learner simply by splitting the training set into two parts and throwing away the labels of one of them.

Let $h_{\mathsf{priv}}$ denote the mapping defined by $\mathcal{A}_{\mathsf{binClas}}$ (Algorithm 2) on a single query (unlabeled data point). That is, for $x \in \mathcal{X}$, $h_{\mathsf{priv}}(x) \in \{0, 1, \bot\}$ denotes the output of $\mathcal{A}_{\mathsf{binClas}}$ on a single input query $x$. Note that w.l.o.g., we can view $h_{\mathsf{priv}}$ as a binary classifier by replacing $\bot$ with a uniformly random label in $\{0, 1\}$. Our private learner is described in Algorithm 3 below.

---

**Algorithm 3** $\mathcal{A}_{\mathsf{Priv}}$: Private Learner

---

**Input:** Unlabeled set of $m$ i.i.d. feature vectors: $\mathcal{Q} = \{\tilde{x}_1, \ldots, \tilde{x}_m\}$, oracle access to our private classifier $h_{\mathsf{priv}}$, oracle access to an agnostic PAC learner $\Theta$ for a class $\mathcal{H}$.

1: **for** $t = 1, \ldots, m$ **do**
2: $\quad \hat{y}_t \leftarrow h_{\mathsf{priv}}(\tilde{x}_t)$
3: **Output** $\hat{h} \leftarrow \Theta(\tilde{D})$, where $\tilde{D} = \{(\tilde{x}_1, \hat{y}_1), \ldots, (\tilde{x}_m, \hat{y}_m)\}$

---

Note that since differential privacy is closed under post-processing, $\mathcal{A}_{\mathsf{Priv}}$ is $(\epsilon, \delta)$-DP w.r.t. the original dataset (input to $\mathcal{A}_{\mathsf{binClas}}$). Note also that the mapping $h_{\mathsf{priv}}$ is independent of $\mathcal{Q}$; it only depends on the input training set $D$ (in particular, on $h_1, \ldots, h_k$), and the internal randomness of $\mathcal{A}_{\mathsf{binClas}}$. We now make the following claim about $h_{\mathsf{priv}}$.

**Claim 4.1.** *Let $0 < \beta \leq \alpha < 1$, and $m \geq 4\log(1/\alpha\beta)/\alpha$. Suppose that $\Theta$ in $\mathcal{A}_{\mathsf{binClas}}$ (Algorithm 2) is an $(\alpha, \beta/k, n/k)$-(agnostic) PAC learner for the hypothesis class $\mathcal{H}$. Then, with probability at least $1 - 2\beta$ (over the randomness of the private training set $D$, and the randomness in $\mathcal{A}_{\mathsf{binClas}}$), we have $\mathsf{err}(h_{\mathsf{priv}}; \mathcal{D}) \leq 3\gamma + 7\alpha = O(\gamma + \alpha)$, where $\gamma = \min_{h \in \mathcal{H}} \mathsf{err}(h; \mathcal{D})$.*

*Proof.* The proof largely relies on the proof of Theorem 3.2. First, note that w.p. $\geq 1 - \beta$ (over the randomness of the input dataset $D$), for all $j \in [k]$, we have $\mathsf{err}(h_j; \mathcal{D}) \leq \alpha$. For the remainder of the proof, we will condition on this event. Let $\tilde{x}_1, \ldots, \tilde{x}_m$ be a sequence of i.i.d. domain points, and $\tilde{y}_1, \ldots, \tilde{y}_m$ be the corresponding (unknown) labels. Now, for every $t \in [m]$, define $v_t \triangleq \mathbf{1}\left(|\{j \in [k] : h_j(\tilde{x}_t) \neq \tilde{y}_t\}| > k/3\right)$. Note that since $(\tilde{x}_1, \tilde{y}_1), \ldots, (\tilde{x}_m, \tilde{y}_m)$ are i.i.d., it follows that $v_1, \ldots, v_m$ are i.i.d. (this is true conditioned on the original dataset $D$). As in the proof of Theorem 3.2, we have: $\mathbb{P}_{\tilde{x}_1, \ldots, \tilde{x}_m}\left[\frac{1}{m}\sum_{t=1}^{m} v_t > 3\left(\alpha + \gamma + \sqrt{\frac{\log(m/\beta)}{2m(\alpha+\gamma)}}\right)\right] < \beta$. Hence, for any

$t \in [m]$, we have $\mathbb{E}_{\tilde{x}_t}[v_t] = \mathbb{E}_{\tilde{x}_1,\ldots,\tilde{x}_m}\left[\frac{1}{m}\sum_{t=1}^m v_t\right] < \beta + 3\left(\alpha + \gamma + \sqrt{\frac{\log(m/\beta)}{2m(\alpha+\gamma)}}\right) \leq 7\alpha + 3\gamma$. Let $\bar{v}_t = 1 - v_t$. Using the same technique as in the proof of Theorem 3.2, we can show that w.p. at least $1 - \beta$ over the internal randomness in Algorithm 2, we have $\bar{v}_t = 1 \Rightarrow h_{\mathsf{priv}}(\tilde{x}_t) = \tilde{y}_t$. Hence, conditioned on this event, we have $\mathbb{P}_{\tilde{x}_t}[h_{\mathsf{priv}}(\tilde{x}_t) \neq \tilde{y}_t] \leq \mathbb{P}_{\tilde{x}_t}[v_t = 1] = \mathbb{E}_{\tilde{x}_t}[v_t] \leq 7\alpha + 3\gamma$. $\qquad\square$

We now state and prove the main results of this section. Let $V$ denote the VC-dimension of $\mathcal{H}$.

**Theorem 4.2** (Sample complexity bound in the realizable case)**.** *Let $0 < \beta \leq \alpha < 1$. Let $m$ be such that $\Theta$ is an $(\alpha, \beta, m)$-agnostic PAC learner of $\mathcal{H}$, i.e., $m = O\left(\frac{V + \log(1/\beta)}{\alpha^2}\right)$. Let the parameter $T$ of $\mathcal{A}_{\mathsf{binClas}}$ (Algorithm 2) be set as in Theorem 3.4. There exists $n = \tilde{O}\left(V^{3/2}/\epsilon\,\alpha^{3/2}\right)$ for the size of the private dataset such that, w.p. $\geq 1 - 3\beta$, the output hypothesis $\hat{h}$ of $\mathcal{A}_{\mathsf{Priv}}$ (Algorithm 3) satisfies $\mathsf{err}(\hat{h}; \mathcal{D}) = O(\alpha)$.*

*Proof.* Let $h^* \in \mathcal{H}$ denote the true labeling hypothesis. We will denote the true distribution $\mathcal{D}$ as $(\mathcal{D}_{\mathcal{X}}, h^*)$. Note that since $T$ is set as in Theorem 3.4, and given the value of $m$ in the theorem statement, we get $k = \tilde{O}(V^2/\epsilon^2\,\alpha^2 n)$. Hence, there is a setting $n = \tilde{O}\left(V^{3/2}/\epsilon\,\alpha^{3/2}\right)$ such that $\Theta$ is an $(\alpha, \beta/k, n/k)$-PAC learner for $\mathcal{H}$ (in particular, sample complexity in the realizable case $= n/k = \tilde{O}(V/\alpha)$). Hence, by Claim 4.1, w.p. $\geq 1 - 2\beta$, $\mathsf{err}(h_{\mathsf{priv}}; \mathcal{D}) \leq 7\alpha$. For the remainder of the proof, we will condition on this event. Note that each $(\tilde{x}_t, \tilde{y}_t)$, $t \in [m]$, is drawn independently from $(\mathcal{D}_{\mathcal{X}}, h_{\mathsf{priv}})$. Now, since $\Theta$ is also an $(\alpha, \beta, m)$-agnostic PAC learner for $\mathcal{H}$, w.p. $\geq 1 - \beta$ (over the *new* set $\tilde{D}$), the output hypothesis $\hat{h}$ satisfies

$$\mathsf{err}(\hat{h}; (\mathcal{D}_{\mathcal{X}}, h_{\mathsf{priv}})) - \mathsf{err}(h^*; (\mathcal{D}_{\mathcal{X}}, h_{\mathsf{priv}})) \leq \mathsf{err}(\hat{h}; (\mathcal{D}_{\mathcal{X}}, h_{\mathsf{priv}})) - \min_{h \in \mathcal{H}} \mathsf{err}(h; (\mathcal{D}_{\mathcal{X}}, h_{\mathsf{priv}})) \leq \alpha.$$

Observe that

$$\mathsf{err}(h^*; (\mathcal{D}_{\mathcal{X}}, h_{\mathsf{priv}})) = \mathbb{E}_{x \sim \mathcal{D}_{\mathcal{X}}}[\mathbf{1}(h^*(x) \neq h_{\mathsf{priv}}(x))] = \mathsf{err}(h_{\mathsf{priv}}; (\mathcal{D}_{\mathcal{X}}, h^*)) = \mathsf{err}(h_{\mathsf{priv}}; \mathcal{D}) \leq 7\alpha,$$

where the last inequality follows from Claim 4.1 (with $\gamma = 0$). Hence, $\mathsf{err}(\hat{h}; (\mathcal{D}_{\mathcal{X}}, h_{\mathsf{priv}})) \leq 8\alpha$. Furthermore, observe that

$$\mathsf{err}(\hat{h}; \mathcal{D}) = \mathbb{E}_{x \sim \mathcal{D}_{\mathcal{X}}}\left[\mathbf{1}(\hat{h}(x) \neq h^*(x))\right] \leq \mathbb{E}_{x \sim \mathcal{D}_{\mathcal{X}}}\left[\mathbf{1}(\hat{h}(x) \neq h_{\mathsf{priv}}(x)) + \mathbf{1}(h_{\mathsf{priv}}(x) \neq h^*(x))\right]$$

$$= \mathsf{err}(\hat{h}; (\mathcal{D}_{\mathcal{X}}, h_{\mathsf{priv}})) + \mathsf{err}(h_{\mathsf{priv}}; \mathcal{D}) \leq 15\alpha.$$

Hence, w.p. $\geq 1 - 3\beta$, we have $\mathsf{err}(\hat{h}; \mathcal{D}) \leq 15\alpha$. $\qquad\square$

**Remark 2.** *In Theorem 4.2, if $\Theta$ is an ERM learner, then the value of $m$ can be reduced to $\tilde{O}(V/\alpha)$. Hence, the resulting sample complexity would be $n = \tilde{O}(V^{3/2}/\epsilon\,\alpha)$, saving us a factor of $\frac{1}{\sqrt{\alpha}}$. This is because the disagreement rate in the labels produced by $\mathcal{A}_{\mathsf{binClas}}$ is $\approx \alpha$, and agnostic learning with such a low disagreement rate can be done using $\tilde{O}(V/\alpha)$ if the learner is an ERM [7, Corollary 5.2].*

**Remark 3.** *Our result involves using an agnostic PAC learner $\Theta$. Agnostic PAC learners with optimal sample complexity can be computationally inefficient. One way to give an efficient construction in the realizable case (with a slightly worse sample complexity) is to use a PAC learner (rather than an agnostic one) in $\mathcal{A}_{\mathsf{Priv}}$ with target accuracy $\alpha$ (and hence, $m = \tilde{O}(V/\alpha)$), but then train the PAC learner in $\mathcal{A}_{\mathsf{binClas}}$ towards a target accuracy $1/m$. Hence, the misclassification rate of $\mathcal{A}_{\mathsf{binClas}}$ can be driven to zero. This yields a sample complexity bound $n = \tilde{O}(V^2/\epsilon\,\alpha)$.*

**Theorem 4.3** (Sample complexity bound in the non-realizable case)**.** *Let $0 < \beta \leq \alpha < 1$, and $m = O\left(\frac{V + \log(1/\beta)}{\alpha^2}\right)$. Let $T$ be set as in Theorem 3.5. There exists $n = \tilde{O}\left(V^{3/2}/\epsilon\,\alpha^{5/2}\right)$ such that, w.p. $\geq 1 - 3\beta$, the output hypothesis $\hat{h}$ of (Algorithm 3) satisfies $\mathsf{err}(\hat{h}; \mathcal{D}) = O(\alpha + \gamma)$.*

*Proof.* The proof is similar to the proof of Theorem 4.2. $\qquad\square$

# 5 Discussion

**Implications, and comparison to prior work on label privacy:** Our results also apply to the setting of label-private learning, where the learner is only required to protect the privacy of the labels in the training set. That is, in this setting, all unlabeled features in the training set can be viewed as public information. This is a less restrictive setting than the setting we consider in this paper. In particular, our construction can be directly used as a label-private learner simply by splitting the training set into two parts and discarding the labels in one of them. The above theorems give sample complexity upper bounds that are only a factor of $\tilde{O}\left(\sqrt{V/\alpha}\right)$ worse than the optimal non-private sample complexity bounds. We note, however, that our sample complexity upper bound for the agnostic case has a suboptimal dependency (by a small constant factor) on $\gamma \triangleq \min_{h \in \mathcal{H}} \mathrm{err}(h; \mathcal{D})$.

Label-private learning has been considered before in [10] and [6]. Both works have only considered pure, i.e., $(\epsilon, 0)$, differentially private learners for those settings, and the constructions in both works are white-box, i.e., they do not allow for modular construction based on a black-box access to a non-private learner. The work of [10] gave upper and lower bounds on the sample complexity in terms of the doubling dimension. Their upper bound involves a smoothness condition on the distribution of the features $\mathcal{D}_{\mathcal{X}}$. The work of [6] showed that the sample complexity (of pure differentially label-private learners) can be characterized in terms of the VC dimension. They proved an upper bound on the sample complexity for the realizable case. The bound of [6] is only a factor of $O(1/\alpha)$ worse than the optimal non-private bound for the realizable case.

**Beyond standard PAC learning with binary loss:** In this paper, we used our algorithmic framework to derive sample complexity bounds for the standard (agnostic) PAC model with the binary 0-1 loss. However, it is worth pointing out that our framework is applicable in more general settings. In particular, if a surrogate loss (e.g., hinge loss or logistic loss) is used instead of the binary loss, then our framework can be instantiated with any non-private learner with respect to that loss. That is, our construction does not necessarily require an (agnostic) PAC learner. However, in such case, the accuracy guarantees of our construction will be different from what we have here for the standard PAC model. In particular, in the surrogate loss model, one often needs to invoke some weak assumptions on the data distribution in order to bound the optimization error [25]. One can still provide meaningful accuracy guarantees since our framework allows for transferring the classification error guarantee of the underlying non-private learner to a classification error guarantee for the final private learner.

*Acknowledgement:* The authors would like to thank Vitaly Feldman, and Adam Smith for helpful discussions during the course of this project. In particular, the authors are grateful for Vitaly's ideas about the possible extensions of the results in Section 4, which we outlined in Remarks 2 and 3. This work is supported by NSF grants TRIPODS-1740850, TRIPODS+X-1839317, and IIS-1447700, a grant from the Sloan foundation, and start-up supports from OSU and UC Santa Cruz.

## Footnotes

[4]Detailed argument given in the full version.

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
