[Reviews · NeurIPS 2018]

Reviewer 1



The paper considers a new differentially private learning setting, that receives a collection of unlabeled public data, on top of the labelled private data of interests and assumes that the two data sets are drawn from the same distribution. The proposed technique allows the use of (non-private) agnostic PAC learners as black boxes oracles, which, when combining with and adapts to the structure of the data sets. The idea is summarized below: 1. Do differentially private model-serving in a data-adaptive fashion, through sparse vector’’ technique and subsample-and-aggregate’’. This only handles a finite number of classification queries. 2. Reduction: Use the labeled private data and non-private blackbox learner to differentially privately release labels, so the released model is DP w.r.t. the private data since it is just post-processing. It behaves similarly to 1 using the properties of an agnostic PAC learner, but can now handle an unbounded number of classification queries. The idea is not entirely new but the particular approach with Algorithm 1 and formal theoretical analysis is (See Point 1 in detailed comments). My only concern is that it is hard to compare the results to the standard non-adaptive DP-learning setting without unlabeled data and their respective upper/lower bounds. (See Point 2 in my detailed comments) Other than that, I like the work and I recommend accepting it for NIPS. Detailed comments: 1. The PATE algorithm used in - Papernot et al. in "Semi-supervised Knowledge Transfer for Deep Learning from Private Training Data" (ICLR 2017, https://arxiv.org/abs/1610.05755). is about the same algorithmically, although Papernot et. al. focuses on empirical evaluation. The differentially private model-serving part is closely related to - Dwork, C., & Feldman, V. (2018). Privacy-preserving Prediction. arXiv preprint arXiv:1803.10266. That said, the authors did acknowledge the above works and claim that they are independent. 2. One of the main motivation of the work, according to Line 35-37 is to “characterize … the improvements one can achieve for private learning … ” (with the additional unlabeled data) “compared to the aforementioned setting” (the setting without such data). I could not find a theorem in the paper that separates the two settings in terms of their sample-complexity. Maybe I missed it somewhere? Intuitively, there shouldn’t be much difference, especially in the setting when m is about the same as n. Because one can throw away half of the labels in the standard setting so it becomes the setting here with half the data. The only concerning matter is that it is not clear whether the proposed algorithm actually attains the lower bound in the standard setting because the nature of the guarantees are very different, and the current results need to work for all (agnostic) PAC learners. Is logistic regression is an agnostic PAC learner from the surrogate loss point of view? Post rebuttal comments: - I admit having written something misleading in the review and I apologize. Indeed the setting is easier than the standard private learning, where both features and labels need to be protected. - My comment that the bounds are not compatible still hold, because this paper works with universal learner oracles, while in standard DP learning, the metric of interest is about empirical excess risk (defined by any convex loss, rather than the 0-1 loss here). - The point about "throwing away a random half of the labels" is in fact about the label-private learning setting. What I would like the authors to compare is the proposed approach and the alternative approach, e.g., that by Chaudhuri and Hsu, which directly analyze the label-privacy for either output perturbation or exponential mechanism. The advantage of the "model agnostic" approach is not clear to me. Regardless, this is a much better paper relative to other papers in my stack. I trust that the authors will do a good revision and clarify these matters in the camera-ready version.

Reviewer 2



The paper tackles the sample complexity of a private agnostic PAC learner. The theoretical explanations could be stimulating for various future work. However, the paper ended abbruptly and continueing in the supplementary file is not mandatory. The authors have to restructure their paper such that implications and the final disvussion fit within the length of the contribution.

Reviewer 3



Summary The paper studies differentially private learning algorithms for classification problems. The main results are the followings: 1) The first results consider an easier problem in which the algorithm only needs to answer a fixed number of m classification queries (instead of a classifier) based on the private data. Then, building on the sample-and-aggregation framework and the sparse vector technique, the paper gives a poly-time black-box reduction from non-private learner to a private algorithm that solves the easier problem. 2) Then, using the labels given by the algorithm for the easier problem, we can use a non-private learner to learn a classifier as a post-processing while keeping privacy! The cost of privacy is in general well-bounded, losing an alpha^{-1/2} factor compared with the optimal non-private learner where alpha is the classification error. Comments: - This is a strong paper in my opinion. The problem is well motivated and extensively studied. The standard exponential mechanism solves the problem if computational efficiency is not of concern. Computationally efficient private learners are, however, designed on a problem-by-problem basis in general. That is why the black-box reduction in this paper is a big deal. - The paper is well-written in general. Readers familiar with the private learning literature can easily grasp the key ideas. - The combination of sample-and-aggregation and sparse-vector is, in hindsight, very simple and natural. It is almost embarrassing that the community didn’t find this before. I take it as a plus of the paper that the techniques are simple and natural. - The paper hides the dependence on eps, the privacy parameter, in most of the bounds. I understand that it is usually simply a 1/eps dependence but it would be nice to make the bound precise at least in the technical sections (so that we can compare the bounds with those achieved by problem-specific techniques).